# Pathway selection in the self-assembly of $Rh_4L_4$ coordination squares under kinetic control

Atsushi Okazawa [1,5], Naoki Sanada[2,5], Satoshi Takahashi[2], Hirofumi Sato [3,4] & Shuichi Hiraoka [2✉]

Pathway selection principles in reversible reaction networks such as molecular self-assembly have not been established yet, because achieving kinetic control in reversible reaction networks is more complicated than in irreversible ones. In this study, we discovered that coordination squares consisting of *cis*-protected dinuclear rhodium(II) corner complexes and linear ditopic ligands are assembled under kinetic control, perfectly preventing the corresponding triangles, by modulating their energy landscapes with a weak monotopic carboxylate ligand (2,6-dichlorobenzoate: $dcb^-$) as the leaving ligand. Experimental and numerical approaches revealed the self-assembly pathway where the cyclization step to form the triangular complex is blocked by $dcb^-$. It was also found that one of the molecular squares assembled into a dimeric structure owing to the solvophobic effect, which was characterized by nuclear magnetic resonance spectroscopy and single-crystal X-ray analysis.

[1] Department of Electrical Engineering and Bioscience, Waseda University, Tokyo 169-8555, Japan. [2] Department of Basic Science, Graduate School of Arts and Sciences, The University of Tokyo, Tokyo 153-8902, Japan. [3] Department of Molecular Engineering, Kyoto University, Kyoto 615-8510, Japan. [4] Fukui Institute for Fundamental Chemistry, Kyoto University, Kyoto 606-8103, Japan. [5]These authors contributed equally: Atsushi Okazawa, Naoki Sanada. ✉email: hiraoka-s@g.ecc.u-tokyo.ac.jp

Pathway-dependent molecular self-assembly has often been seen in the realm of nature, especially in the multi-component self-assembly of viruses[1] and proteasomes[2]. One of the reasons why such a sequential assembly process is adopted is because the formation of undesired assemblies whose structures are far from the desired one is efficiently prevented in the assembly process. If there exist some other assemblies with similar thermodynamic stabilities, selective assembly of the desired one under thermodynamic control is impossible. In such cases, pathway selection under kinetic control is inevitable, but the general principles of pathway selection in complicated reversible reaction networks have yet to be fully understood[3].

Molecular square is one of the most representative molecular architectures in coordination self-assembly[4–24]. Thanks to the L–M–L bond angle of 90° around transition metal ions with octahedral and square planer coordination spheres, upon complexation with linear ditopic ligands, square complexes are expected to be thermodynamically most stable over other polygonal structures. However, in many cases triangular complexes were coproduced in solution even using rigid linear ditopic ligands as the side of the polygons, although triangles are geometrically largely different from squares[25–45]. Therefore, the selective formation of molecular squares and the understanding of their origin remain challenging in supramolecular chemistry[46].

Under the condition where the energy barriers between square and triangular compounds are relatively high, there is a chance to produce the square in a higher yield than that obtained at equilibrium, kinetically preventing the triangle. Such a kinetic approach is quite limited in artificial molecular self-assembly, except for unexpected observations of kinetic traps[47–55]. Our understanding of the energy landscapes of molecular self-assembly is not as thorough as we can rationally design them, and to the best of our knowledge, there is no report on the selective formation of a triangular or square complex under kinetic control.

Here, we report the self-assembly of the $Rh_4L_4$ squares in solution under kinetic control, perfectly preventing the formation of the $Rh_3L_3$ triangle using a designed monotopic carboxylate as the leaving ligand (dcb⁻ in Fig. 1). Experimental and numerical analyses of the self-assembly mechanism indicate that the free energy landscape of the self-assembly was properly modulated so that the pathways to the triangular complex were blocked. Furthermore, the monocarboxylate facilitates ligand exchanges,

resulting in the conversion of the kinetically trapped triangles into the squares under mild condition. It was also found that one of the $Rh_4L_4$ squares aggregates to form a structurally well-defined dimer in solution by the solvophobic effect, owing to the electronically neutral nature of the Rh(II) complexes.

## Results and discussion

### Self-assembly of Rh(II)-triangle and square with CH₃CN as the leaving ligand.
Cotton and his co-workers and others reported that molecular squares composed of cis-protected dinuclear Rh(II) complexes ($Rh^{2+}$: $[Rh_2(DAniF)_2]^{2+[56]}$ and $[Rh_2(O_2C-R-CO_2)]^{2+[57]}$) and linear dicarboxylates ($L^{2-}$) were selectively obtained in crystalline states[56,57]. As Rh(II)–carboxylate coordination bonds in dinuclear Rh(II) complexes are relatively strong, the reversibility of the coordination bond is low[58], which would allow the self-assembly of the $Rh_4L_4$ squares to proceed under kinetic control. Thus, we were interested in why the $Rh_4L_4$ squares were selectively produced in the solid state in the previous research.

We first performed the self-assembly from $[Rh(CH_3CN)_4]$ $(BF_4)_2$ and $1^{2-}$ in CH₃CN ($[1^{2-}]$ = 10 mM) according to the literature[56]. Dark red solid precipitated in 1 day. Its ¹H NMR spectrum in CDCl₃ showed two prominent singlet signals of $H^d$ (aromatic protons of $1^{2-}$ as shown in Fig. 2) in a 1:3 integral ratio, suggesting the formation of two highly symmetric cyclic structures (Supplementary Fig. 1). Crystallization of this mixture in CHCl₃ gave single crystals of $Rh_41_4$ as was reported in the literature[56]. The major ¹H NMR signals are the same as those of $Rh_41_4$ (Supplementary Fig. 2), so the minor signals are expected to correspond to the $Rh_31_3$ triangle, which is supported by ¹H DOSY spectroscopy (Supplementary Fig. 3), ESI-TOF mass spectrometry (Supplementary Fig. 4a), and other experiments (details are shown in Supplementary Fig. 2). Therefore, the $Rh_41_4$ square and the $Rh_31_3$ triangle were produced in CH₃CN.

To investigate the solution process, self-assembly was performed in CDCl₃ at 298 K ($[1^{2-}]$ = 1.0 mM), where precipitation did not occur during the self-assembly, resulting in a mixture of $Rh_31_3$ and $Rh_41_4$ in a 1:2 ratio (Fig. 2a). Similarly, self-assembly of $[Rh(CH_3CN)_4](BF_4)_2$ and another ditopic ligand ($2^{2-}$) in CDCl₃ gave the triangular and square complexes in a 1:2 ratio (Supplementary Figs. 2, 4c, 5, and 6). Therefore, the self-assembly of $[Rh(CH_3CN)_4]^{2+}$ and the linear ditopic ligands ($L^{2-}$)

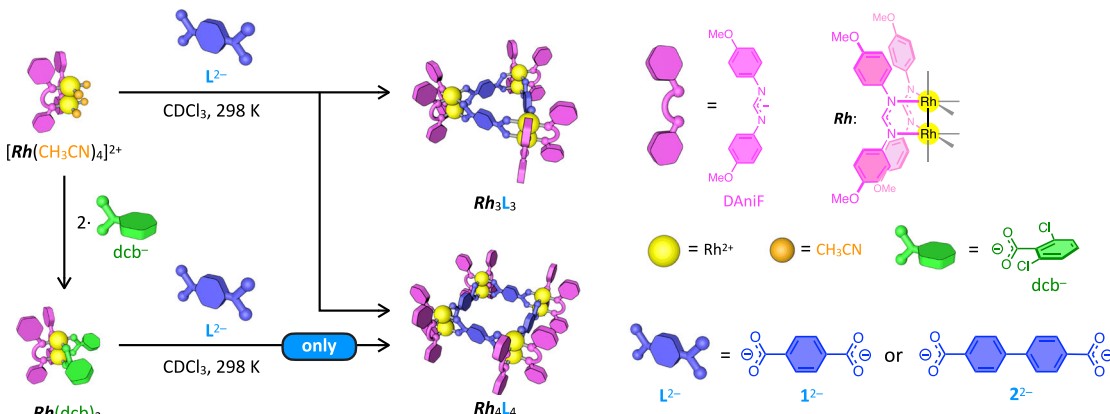

**Fig. 1 Self-assembly of the Rh₄L₄ squares in this study.** Schematic representation of kinetically controlled self-assembly of the $Rh_4L_4$ squares from cis-protected dinuclear Rh(II) complex ($Rh^{2+}$: $[Rh_2(DAniF)_2]^{2+}$) and linear dicarboxylate ligand ($L^{2-}$: $1^{2-}$ or $2^{2-}$) in solution. When the metal source is $[Rh(CH_3CN)_4]^{2+}$, where CH₃CN is the leaving ligand, $Rh_3L_3$ triangle and $Rh_4L_4$ square complexes were produced in a 1:2 ratio. In contrast, when the self-assembly was carried out using $Rh(dcb)_2$ as the metal source (dcb⁻: 2,6-dichlorobenzoate), dcb⁻ modulates the energy landscape of the self-assembly so that the formation of the $Rh_3L_3$ triangle is prevented, producing the $Rh_4L_4$ square only. The axial ligands on the dinuclear Rh(II) centers are omitted for clarity.

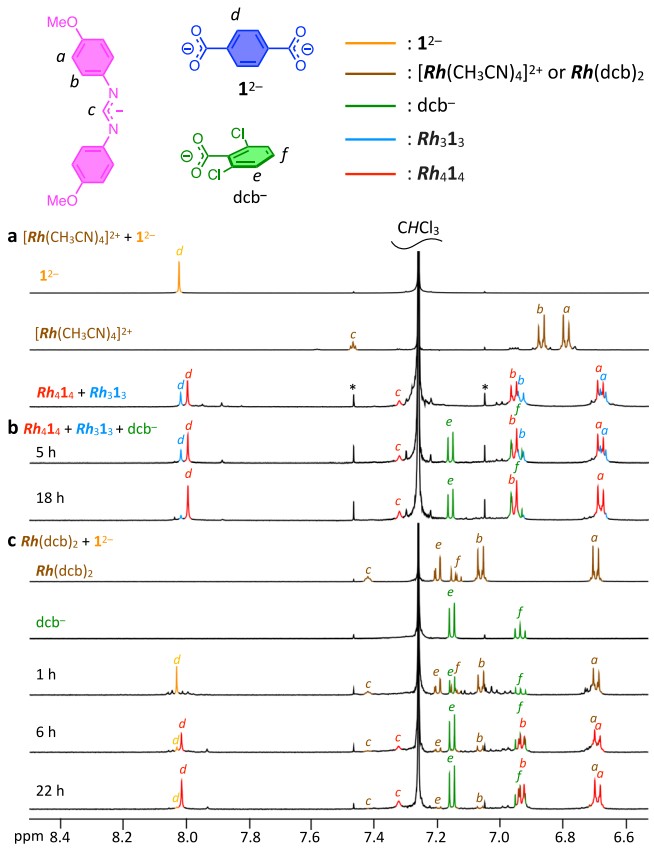

**Fig. 2 $^1$H NMR spectra of the self-assembly of the $Rh_4\mathbf{1}_4$ square.** $^1$H NMR spectra (500 MHz, CDCl$_3$, 298 K, aromatic region) of the self-assembly of the $Rh_4\mathbf{1}_4$ squares under various conditions ([$\mathbf{1}^{2-}$] = [$Rh^{2+}$] = 1 mM). **a** Self-assembly from [$Rh$(CH$_3$CN)$_4$](BF$_4$)$_2$ and $\mathbf{1}^{2-}$ in CDCl$_3$ at 298 K, giving the $Rh_3\mathbf{1}_3$ triangle and the $Rh_4\mathbf{1}_4$ square in 23% and 39% yields, respectively. Asterisks indicate the carbon satellite of CHCl$_3$. **b** Addition of $n$-Bu$_4$N·dcb in a mixture of the $Rh_3\mathbf{1}_3$ triangle and the $Rh_4\mathbf{1}_4$ square obtained from the self-assembly of [$Rh$(CH$_3$CN)$_4$](BF$_4$)$_2$ and $\mathbf{1}^{2-}$ in CDCl$_3$ after convergence. The $Rh_3\mathbf{1}_3$ triangle was converted into the $Rh_4\mathbf{1}_4$ square at 298 K assisted by dcb$^-$, although heating at 100 °C for 2 days is necessary without dcb$^-$ (Supplementary Fig. 14). **c** Self-assembly of the $Rh_4\mathbf{1}_4$ square from $Rh$(dcb)$_2$ and $\mathbf{1}^{2-}$ in CDCl$_3$ at 298 K to produce the $Rh_4\mathbf{1}_4$ square in a 65% yield without formation of the $Rh_3\mathbf{1}_3$ triangle during self-assembly. The yields were determined based on the internal standard.

at room temperature gave a mixture of a molecular square (major) and a triangle (minor) in solution, and the $Rh_4\mathbf{1}_4$ square was selectively crystallized from the mixture.

**Kinetic self-assembly of Rh(II)-squares.** We then investigated the kinetic effect of the leaving ligand on $Rh$-based self-assembly. 2,6-Dichlorobenzoate (dcb$^-$) was chosen as the leaving ligand (Fig. 1). Because the p$K_a$ value of Hdcb in H$_2$O (1.82)[59,60] is lower than that of benzoic acid (4.21), the coordination ability of dcb$^-$ is lower than those of the ditopic ligands (L$^{2-}$), suggesting that dcb$^-$ can act as a leaving ligand. To test the potential of dcb$^-$ as a leaving ligand, the ligand exchange of dcb$^-$ in $Rh$(dcb)$_2$ with $p$-toluate (tol$^-$) in CDCl$_3$ was monitored by $^1$H NMR spectroscopy (Supplementary Fig. 7a). The ligand exchange proceeded at 298 K to produce $Rh$(tol)$_2$ in 82% yield (Supplementary Fig. 7b), indicating that dcb$^-$ can be used as a leaving ligand in $Rh$-based self-assembly at room temperature.

The self-assembly of $Rh$(dcb)$_2$ and L$^{2-}$ ($\mathbf{1}^{2-}$ and $\mathbf{2}^{2-}$) in CDCl$_3$ at 298 K was monitored by $^1$H NMR spectroscopy (Fig. 2c and

Supplementary Fig. 8). Surprisingly, the $^1$H NMR signals of the $Rh_3$L$_3$ triangle did not appear at all during the self-assembly for $\mathbf{1}^{2-}$ and $\mathbf{2}^{2-}$. These results indicate that the energy landscape of the self-assembly was dramatically altered by dcb$^-$, kinetically producing only the $Rh_4$L$_4$ squares.

**Self-assembly mechanism of Rh(II)-squares.** We were interested in why the $Rh_4$L$_4$ squares were selectively produced with dcb$^-$ as the leaving ligand under kinetic control, so the self-assembly processes were investigated by QASAP (quantitative analysis of self-assembly process)[61–63]. Basically, it is difficult to obtain information about intermediates in molecular self-assembly, because most of intermediates are unobservable. Indeed, only small minor signals for intermediates were observed by $^1$H NMR spectroscopy during the self-assembly of the $Rh_4$L$_4$ squares (Fig. 2c and Supplementary Fig. 8). In such cases, QASAP provides valuable insight into the self-assembly pathway from time-development of the average composition of all intermediates even when the intermediates cannot be observed. We have applied QASAP to Pd(II)- and Pt(II)-based coordination self-assemblies so far to determine the major self-assembly pathway(s) and the rate-determining step. QASAP of the $Rh_4$L$_4$ squares is the first example of QASAP for metal-organic assemblies with transition metals other than Pd(II) and Pt(II) ions.

The substrates ($Rh$(dcb)$_2$ and L$^{2-}$) and the products ($Rh_4$L$_4$ and dcb$^-$) were quantified by $^1$H NMR spectroscopy (Fig. 3c), and the change in the average composition of all intermediates with time was plotted in a 2D map ($n$-$k$ map) (Fig. 3b). The species regarding the self-assembly are expressed by $Rh_a$L$_b$(dcb)$_c$ ($a$–$c$ are 0 or positive integer), which is indicated as ($a$,$b$,$c$) for simplicity. The ($n$, $k$) values of $Rh_a$L$_b$(dcb)$_c$ are defined as $n = (2a - c)/b$ and $k = a/b$[61–63]. The $n$ value indicates the average number of $Rh^{2+}$ bound to a single ditopic ligand, L$^{2-}$. The $k$ value indicates the ratio of $Rh^{2+}$ against L$^{2-}$. The experimentally obtained ($n$, $k$) values, which is indicated as ($\langle n \rangle$, $\langle k \rangle$), are calculated for the average composition of all intermediates ($Rh_{\langle a \rangle}$L$_{\langle b \rangle}$(dcb)$_{\langle c \rangle}$), whose time-development was used for the discussion on the self-assembly process (Fig. 3b).

There are three types of chain intermediates classified according to the difference in the terminals (Types I–III) (Fig. 3a)[64,65]. The ($n$, $k$) values of the three types of oligomers are distinctly plotted on different straight lines in the $n$-$k$ map (Fig. 3b). With an increase in the degree of oligomerization ($m$ in Fig. 3a), the ($n$, $k$) values become close to (2, 1), which is the ($n$, $k$) value of the cyclic structures (triangle and square). Thus, the ($n$, $k$) plot enables us to discuss which type of oligomers was mainly produced and how large the chain-like oligomers grew during self-assembly.

With regard to QASAP for the $Rh_4\mathbf{1}_4$ square, the $\langle n \rangle$ value was smaller than 1 at an early stage (Supplementary Fig. 9 and Supplementary Table 1), which suggests that $Rh^{2+}$ units in the intermediates have more than two carboxylate ligands ($\mathbf{1}^{2-}$ and/or dcb$^-$), so further analysis could not be performed (for detailed discussion, see Supplementary Fig. 9). Such a strange behavior was not found for the self-assembly of the $Rh_4\mathbf{2}_4$ square (Fig. 3b and Supplementary Fig. 10 and Supplementary Table 2). The $\langle n \rangle$ value increased with an almost constant $\langle k \rangle$ value of 1, indicating the formation of Type II oligomers as main intermediates, whose $m$ value finally reached 3 or 4, corresponding to 4 or 5 $Rh^{2+}$ units in the oligomers.

**Numerical analysis of self-assembly pathway.** The major self-assembly pathway was determined by numerical analysis (NASAP: numerical analysis of self-assembly process)[66]. In NASAP, the QASAP data were numerically analyzed based on the

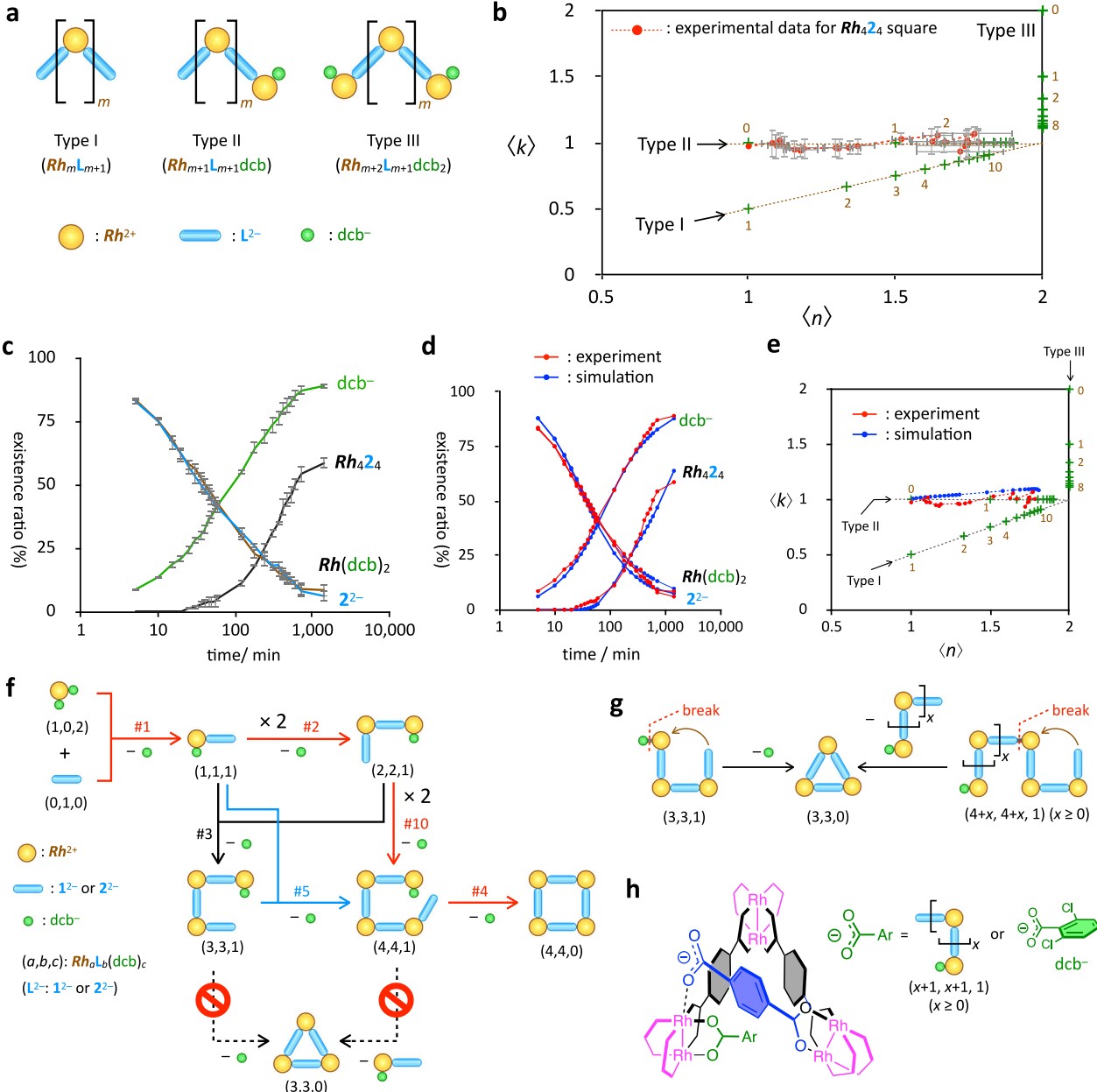

**Fig. 3 Self-assembly mechanism of the $Rh_4 2_4$ square under kinetic control. a** Three types of possible chain intermediates in the self-assembly of $M_4L_4$ square. Type I: $M_mL_{m+1}$, Type II: $M_{m+1}L_{m+1}(dcb)$, Type III: $M_{m+2}L_{m+1}(dcb)_2$. **b** Plots of the existence ratios of the substrates and products in the self-assembly of the $Rh_4 2_4$ square from $Rh(dcb)_2$ and $2^{2-}$ in $CDCl_3$ at 298 K. $[Rh]_0 = [2^{2-}]_0 = 0.86$ mM. **c** Plots of the $(\langle n \rangle, \langle k \rangle)$ values in the n-k map of the $Rh_4 2_4$ square (red filled circles). Green crosshairs indicate the $(n, k)$ values of the chain intermediates. The three types of chain intermediates, Types I, II, and III, are plotted on each straight line. The brown number indicates $m$ in each type of oligomer in **a**. The definitions of $\langle n \rangle$ and $\langle k \rangle$ values are shown in the main text (Eqs. 1 and 2). The data in (**b**) and (**c**) are shown as the average of the three runs of QASAP with standard errors. **d** Comparison of the existence ratio of the substrates and the products between QASAP (red) and NASAP (blue). **e** Comparison of the n-k plot between QASAP (red) and NASAP (blue). Both (**d**) and (**e**) indicate that the numerical simulation results reproduce the experimental counterparts well. **f** Dominant self-assembly pathways of the $Rh_4 2_4$ square. Red arrows indicate the major self-assembly pathway. $(a,b,c)$ indicates $Rh_a 2_b(dcb)_c$. The numbers above the reaction arrows indicate the elementary reactions with high net reaction frequency listed in Supplementary Table 4. Longer Type II oligomers than (4,4,1), such as (5,5,1) and (6,6,1), are produced during the self-assembly, although they are not involved in the major self-assembly pathway (Supplementary Fig. 13). **g** Two possible pathways to produce triangle (3,3,0) from Type II intermediate (3,3,1) (left) and from Type II oligomers with more than three $Rh^{2+}$ units $(4+x, 4+x, 1)$ $(x \geq 0)$ (right). The green sphere indicates the leaving ligand(s) (dcb⁻ or two molecules of $CH_3CN$). **h** A plausible key structure in the triangular formation through associative ligand exchange process. Steric repulsion caused by $ArCO_2^-$ (in green) in the cyclic intermediate and transition state of the triangular formation process would prevent the cyclization.

reaction network model where possible intermediates and their elementary reactions were considered. We prepared a reaction network model consisting of 34 species with a maximum number of $Rh^{2+}$ of 10 and 1694 elementary reactions between them

(Supplementary Fig. 11). The elementary reactions in the network were classified into 10 groups according to the reaction type (Supplementary Fig. 12). Then, the forward and backward rate constants for the elementary reactions were used as variable

parameters to fit the QASAP data. No good dataset was found unless the rate constants of the triangular formations ($k_8$ and $k_9$) were set to almost zero, which is consistent with the experimental result that the [1]H NMR signals of $Rh_3 2_3$ did not appear during the self-assembly (Supplementary Fig. 8). A set of rate constants that reproduced the experimental results well (Fig. 3d, e) was found (Supplementary Table 3).

With the dataset of the rate constants thus obtained, the self-assembly of $Rh_4 2_4$ was numerically simulated in the reaction network, which enabled us to determine the major self-assembly pathway. The top 10 elementary reactions with high net frequencies are listed in Supplementary Table 4. Net frequency represents the difference between the occurrence in the forward direction and that in the backward direction for an elementary reaction; thus, the net frequency indicates the actual degree of progress of the elementary reaction[67]. Connecting the elementary reactions with the highest net reaction frequency from $Rh_4 2_4$ to the substrates resulted in the dominant self-assembly pathways (Fig. 3f). As expected from the n-k analysis, $Rh_4 2_4$ was assembled, producing Type II species as the major intermediates, and the largest intermediate in the major pathway is $Rh_4 2_4(dcb)_1$, which is indicated as (4,4,1) (Fig. 3f).

The change in the existence ratios of the intermediates in the major self-assembly pathway shows that the existence of (4,4,1) is much smaller than that of the other intermediates (Supplementary Fig. 13), indicating that the cyclization of (4,4,1) occurs quickly and that the formation of (4,4,1) from (2,2,1) and from (3,3,1) is the rate-determining steps (Fig. 3f)[68].

**Reason for the selective formation of Rh(II)-squares under kinetic control**. Next, we discuss why the pathways to the $Rh_3 L_3$ triangles were blocked with dcb⁻ as the leaving ligand. There are two types of reactions that produce $Rh_3 L_3$ by the cyclization of Type II intermediates: (i) the cyclization of (3,3,1) to form (3,3,0) by breaking the coordination bond between Rh(II) and the leaving ligand (CH₃CN or dcb⁻) (Fig. 3g, left) and (ii) the cyclization of oligomers longer than (3,3,1), (4+x, 4+x, 1) (x ≥ 0), by breaking a Rh–L coordination bond (Fig. 3g, right). Note that the latter reaction is not affected by the leaving ligand. Therefore, the formation of $Rh_3 L_3$ triangles from (4+x, 4+x, 1) (x ≥ 0) should be prevented either using $Rh(dcb)_2$ or $[Rh(CH_3CN)_4]^{2+}$. In addition, when dcb⁻ is the leaving ligand, the cyclization of (3,3,1) to form (3,3,0) should be prevented. In other words, the cyclization of $Rh_3 L_3(CH_3CN)_2$ occurs, while that of $Rh_3 L_3(dcb)$ is kinetically blocked.

Considering the associative ligand exchange mechanism of Rh(II)–carboxylate bonds[69,70], triangular formation should take place by the attack of the terminal non-coordinated carboxylate group of L to the axial site of a Rh(II) center in Type II intermediates (Fig. 3h). The instability of a resultant cyclic intermediate or a transition state with dcb⁻ would be the reason for the suppression of the cyclization of $Rh_3 L_3(dcb)$.

**Conversion of Rh(II)-triangles into Rh(II)-squares assisted by dcb⁻**. The conversion between the triangular and square complexes was very slow due to the inertness of Rh–carboxylate bonds. Heating a 1:2 mixture of $Rh_3 1_3$ and $Rh_4 1_4$ in CDCl₃ at 60 °C did not show any change of their ratio (Supplementary Fig. 14a). The conversion of $Rh_3 1_3$ into $Rh_4 1_4$ was realized by heating the mixture in dimethylacetamide (DMA) at 100 °C for 2 days (Supplementary Fig. 14b).

Considering the associative ligand exchange mechanism of Rh(II)-carboxylate bonds, the ligand exchange is expected to be facilitated by monotopic carboxylate with weak coordination ability such as dcb⁻. Thus, n-Bu₄N·dcb was added to a mixture of

$Rh_3 1_3$ and $Rh_4 1_4$ in CDCl₃, and the reaction at 298 K was monitored by [1]H NMR spectroscopy (Fig. 2b and Supplementary Table 5). The signals assigned to $Rh_3 1_3$ slowly decreased with time and almost disappeared after 18 h. The conversion of $Rh_3 1_3$ into $Rh_4 1_4$ was also confirmed by ESI-TOF mass spectrometry (Supplementary Fig. 4b). Likewise, the conversion of $Rh_3 2_3$ into $Rh_4 2_4$ also occurred at 298 K in 13 h with dcb⁻ (Supplementary Figs. 4d and 15 and Supplementary Table 6). These results indicate that dcb⁻ greatly decreased the energy barriers of the conversion of $Rh_3 L_3$ triangles to $Rh_4 L_4$ squares as a catalyst. However, the time scale of the conversion of the triangle into the square using the catalyst is much slower than that of the assembly of the square from the substrates ($Rh_2(dcb)_2$ and the ditopic ligand). The idea that the selective self-assembly of the $Rh_4 L_4$ squares took place by the conversion of the $Rh_3 L_3$ triangles into the $Rh_4 L_4$ squares by the catalytic effect of dcb⁻ during the self-assembly is ruled out by the fact that the signals of the $Rh_3 L_3$ triangles were not observed during the self-assembly (Fig. 2c and Supplementary Fig. 8).

**Supramolecular dimerization of Rh(II)-square by solvophobic effect**. The dinuclear Rh(II)-based assemblies are neutral molecules, which is largely different from other positively or negatively charged coordination assemblies with counter ions. One of the advantages of neutral coordination assemblies is that they can be assembled into higher order structure without electrostatic repulsion between themselves, leading to a closely packed structure in the crystalline state and in solution. A new set of signals appeared when $Rh_4 1_4$ was dissolved in CD₃NO₂/CDCl₃ (9:1 (v/v)) (Fig. 4c). A similar spectral change was also observed in acetone-$d_6$ and CD₃CN with 10 volume% of CDCl₃ (Supplementary Fig. 16). A significant up-field shift of aromatic protons (H$^a$ and H$^b$) suggests the aggregation of the square (Fig. 4a). All signals except for H$^c$ were observed as two sets with a 1:1 integral ratio, which is consistent with the crystal structure of the $[Rh_4 1_4(dmso-S)_4]_2$ dimer (Fig. 4b) obtained from acetone/DMSO.

The $(Rh_4 1_4)_2$ dimer in CD₃NO₂/CDCl₃ (9:1 (v/v)) was characterized by [1]H DOSY, (H, H)-COSY, and (H, H)-NOESY spectroscopy (Fig. 4d and Supplementary Figs. 17 and 18). A cross peak was found between H$^{d(out)}$ and H$^{d(in)}$ protons in the NOESY spectrum (Supplementary Fig. 18), indicating the chemical exchange of the two proton signals caused by the rotation of the 1,4-phenylene ring in $1^{2-}$, which would be the reason for the broadening of the two H$^d$ signals (Fig. 4c). The equilibrium of the monomer ($Rh_4 1_4$) and the dimer (($Rh_4 1_4)_2$) shifted toward the dimer with an increase in the composition ratio of CD₃NO₂/CDCl₃ (Supplementary Fig. 19), indicating the dimerization of the $Rh_4 1_4$ square due to the solvophobic effect. The [1]H NMR signals of the monomer did not appear even at a low concentration of $[Rh_4 1_4]_0 = 4.6$ μM (Supplementary Fig. 20). Further dilution caused the NMR spectrum to significantly broaden with a low S/N ratio. According to these results, the dimerization constant for $(Rh_4 1_4)_2$ is estimated to be higher than $10^7 M^{-1}$ under the assumption that 10% monomer exists (Supplementary Fig. 20).

## Conclusions

In conclusion, the self-assembly of the $Rh_4 L_4$ squares was kinetically controlled to produce $Rh_4 L_4$ squares selectively assisted by monotopic carboxylate ligand (dcb⁻), which prevents the cyclization of the $Rh_3 L_3(dcb)$ chain intermediate. This is the first example of the selective formation of coordination squares under kinetic control in solution. The metastable yet kinetically highly stabilized $Rh_3 L_3$ triangles can be transformed into the $Rh_4 L_4$ squares with dcb⁻ under very mild conditions. These results

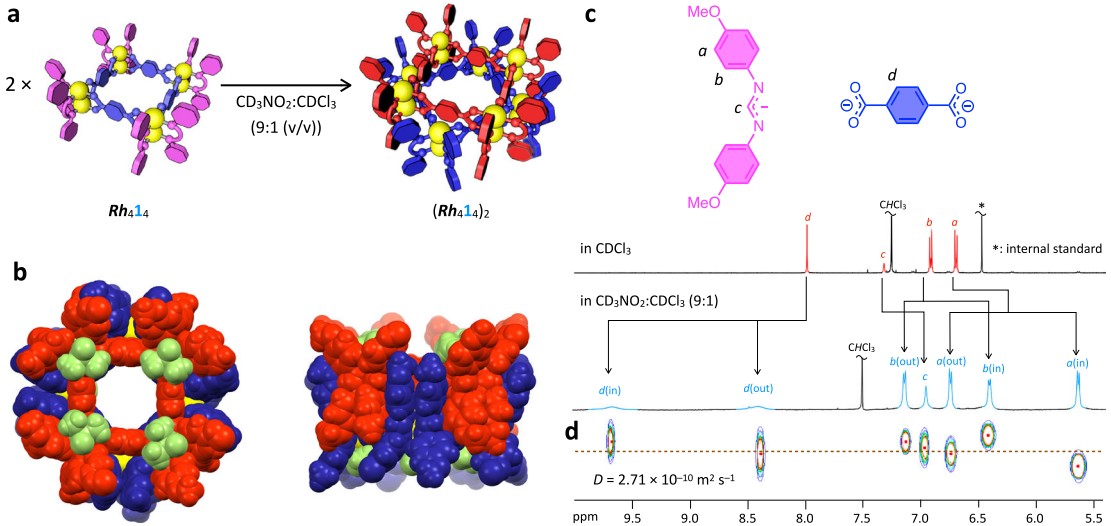

**Fig. 4 Dimerization of the $Rh_4 1_4$ square in solution. a** Supramolecular dimer formation of the $Rh_4 1_4$ square by the solvophobic effect. **b** Crystal structure of [$Rh_4 1_4$(dmso-S)$_4$]$_2$. Two $Rh_4 1_4$ squares engaged each other are shown in red and blue. DMSO molecules axially coordinating to the Rh(II) centers are colored in green. **c** $^1$H NMR spectra (500 MHz, 298 K, aromatic region) of ($Rh_4 1_4$)$_2$ in CD$_3$NO$_2$/CDCl$_3$ (9:1 (v/v)) and $Rh_4 1_4$ in CDCl$_3$. **d** $^1$H DOSY spectrum of ($Rh_4 1_4$)$_2$ in CD$_3$NO$_2$/CDCl$_3$ (9:1 (v/v)).

indicate that the energy landscape of the dinuclear Rh(II)-based coordination self-assembly can be tuned by the leaving ligand. This basic knowledge can be applied to a wide range of Rh(II)-based metal-organic polyhedra (MOPs)[71–73], whose self-assembly is limited compared with MOPs composed of other transition metal ions owing to the relatively inert equatorial Rh(II)–carboxylate bonds. Because Rh(II)-based MOPs possess high thermal and chemical stabilities and catalytic activity, they are expected as a platform for new materials, so exploring the novel structures of Rh(II)-based MOPs and their efficient self-assembly by modulating the energy landscapes will make a great progress in this field.

## Methods

**General information.** $^1$H NMR spectra were recorded using a Bruker AV-500 (500 MHz) spectrometer. All $^1$H NMR spectra were referenced using a residual solvent peak, CDCl$_3$ ($\delta$ 7.26), acetone-$d_6$ ($\delta$ 2.05), CD$_3$NO$_2$ ($\delta$ 4.33). Electrospray ionization time-of-flight (ESI-TOF) mass spectra were obtained using a Waters Xevo G2-S Tof mass spectrometer.

**Materials.** Unless otherwise noted, all solvents and reagents were obtained from commercial suppliers (TCI Co., Ltd., WAKO Pure Chemical Industries Ltd., KANTO Chemical Co., Inc., and Sigma-Aldrich Co.) and were used as received. Deuterated solvents were used after dehydration with Molecular Sieves 4 Å. [$Rh$(CH$_3$CN)$_4$](BF$_4$)$_2$ ($Rh$: Rh$_2$(DAniF)$_2$, DAniF: $N$,$N'$-bis(4-methoxyphenyl)formamidinate)[74] was prepared according to the literature. The $^1$H and $^{13}$C{$^1$H} NMR spectra of all substrates are available in Supplementary Data 1. $Rh$(dcb)$_2$ was prepared by mixing [$Rh$(CH$_3$CN)$_4$](BF$_4$)$_2$ and $n$-Bu$_4$N·dcb in a 1:2 ration in NMR tube at 298 K and were used for the self-assembly after the quantification based on the internal standard ([2.2]paracyclophane). See more detail in section "Monitoring of the self-assembly of the $Rh_4 L_4$ squares".

## QASAP of the $Rh_4 L_4$ squares

*Monitoring of the self-assembly of the $Rh_4 L_4$ squares.* A 2.4 mM solution of [2.2]paracyclophane in CHCl$_3$ (125 μL), which was used as an internal standard, was added to two NMR tubes (tubes **I** and **II**) and the solvent was removed in vacuo. A CDCl$_3$

solution of $Rh$(dcb)$_2$ was prepared as solution **A** (10 mM). Solution **A** (50 μL) and CDCl$_3$ (400 μL) were added to tube **I**. The exact concentration of $Rh$(dcb)$_2$ in solution **A** was determined through the comparison of the signal intensity with [2.2]para-cyclophane by $^1$H NMR. A solution of ditopic ligand ($n$-Bu$_4$N)$_2$·L (L = $1^{2-}$ or $2^{2-}$; 2.5 mM) in CDCl$_3$ (200 μL) and CDCl$_3$ (250 μL) were added to tube **II** and the exact amount of L$^{2-}$ in tube **II** was determined through the comparison of the signal intensity with [2.2]paracyclophane by $^1$H NMR. Then, 1.0 eq. (against the amount of ligand L$^{2-}$ in tube **II**) of solution **A** (*ca.* 50 μL; the exact amount was determined based on the exact concentrations of solution **A** and of L$^{2-}$ in tube **II**) was added to tube **II**. The self-assembly of the $Rh_4 L_4$ square was monitored at 298 K by $^1$H NMR spectroscopy. The selected $^1$H NMR spectra measure during the self-assembly are shown in Fig. 2c and Supplementary Fig. 8 ([$1^{2-}$]$_0$ = 1.0 mM and [$2^{2-}$]$_0$ = 0.86 mM). The amounts and the existence ratios of L$^{2-}$, $Rh$(dcb)$_2$, the $Rh_4 L_4$ square, and dcb$^-$ were quantified by the integral value of each $^1$H NMR signal against the signal of the internal standard ([2.2]paracyclophane) according to the determination methodology described in the section "Determination of the existence ratios of each species". The data, the average values of the existence ratios, and the ($\langle n \rangle$, $\langle k \rangle$) values with standard errors are listed in Supplementary Tables 1 and 2 for the self-assembly of the $Rh_4 1_4$ and $Rh_4 2_4$ squares, respectively.

*Determination of the existence ratio of each species.* The relative integral value of each $^1$H NMR signal against the internal standard [2.2]paracyclophane is used as the integral value in this description. We define the integral values of the signal for the substrates and the products at each time $t$ as follows:

*QASAP for the $Rh_4 1_4$ square.* $I_L(t)$: 1/4 of the integral value of the $d$ proton in free ligand $1^{2-}$
$I_M(t)$: 1/2 of the integral value of the $e$ proton of dcb$^-$ in $Rh$(dcb)$_2$
$I_{440}(t)$: 1/4 of the integral value of the $d$ proton in the $Rh_4 1_4$ square
$I_{dcb}(t)$: the integral value of the $f$ proton of free dcb$^-$ (before 1 h) or 1/2 of the integral value of the $e$ proton of free dcb$^-$ (after 1 h)

*QASAP for the $Rh_42_4$ square.* $I_L(t)$: 1/4 of the integral value of the *e* proton in free ligand $2^{2-}$

$I_M(t)$: 1/2 of the integral value of the *f* proton of dcb⁻ in $Rh(dcb)_2$

$I_{440}(t)$: 1/4 of the integral value of the *d* proton in the $Rh_42_4$ square

$I_{dcb}(t)$: the integral value of the *g* proton of free dcb⁻

$I_M(0)$ was determined based on the exact concentration of solution **A** determined by ¹H NMR and the exact volume of solution **A** added into tube **II**.

$I_L(0)$ was determined by ¹H NMR measurement before the addition of solution **A** into tube **II**.

**Existence ratio of $Rh(dcb)_2$.** As the total (initial) amount of $Rh(dcb)_2$ corresponds to $I_M(0)$, the existence ratio of $Rh(dcb)_2$ at *t* is expressed by $I_M(t)/I_M(0)$.

**Existence ratio of $L^{2-}$.** As the total amount of free ligand $L^{2-}$ corresponds to $I_L(0)$, the existence ratio of $L^{2-}$ at *t* is expressed by $I_L(t)/I_L(0)$.

**Existence ratio of the $Rh_4L_4$ square.** As the total amount of the $Rh_4L_4$ square is quantified based on $L^{2-}$, the existence ratio of the $Rh_4L_4$ square at *t* is expressed by $I_{440}(t)/I_L(0)$.

**Existence ratio of dcb⁻.** As the total amount of dcb⁻ corresponds to $I_M(0)$, the existence ratio of dcb⁻ at *t* is expressed by $I_{dcb}(t)/I_M(0)$.

**Existence ratio of the total intermediates not observed by ¹H NMR (Int).** The existence ratio of the total intermediates not observed by ¹H NMR (Int) is determined based on the amount of $L^{2-}$ in Int. Thus, the existence ratio of Int is calculated by subtracting the other species containing $L^{2-}$ (free $L^{2-}$, and the $Rh_4L_4$ square) from the total amount of $L^{2-}$ ($I_L(0)$). The existence ratio of Int at *t* is expressed by $(I_L(0) - I_L(t) - I_{440}(t))/I_L(0)$.

$\langle a \rangle$

The initial amount of $Rh$ centers corresponds to $I_M(0)/2$.

The amount of $Rh$ centers in $Rh(dcb)_2$ at *t* corresponds to $I_M(t)/2$.

The amount of $Rh$ centers in the $Rh_4L_4$ square at *t* corresponds to $I_{440}(t)$.

The amount of $Rh$ centers in Int at *t* is thus expressed by $I_M(0)/2 - I_M(t)/2 - I_{440}(t)$.

$\langle b \rangle$

The initial amount of $L^{2-}$ corresponds to $I_L(0)$.

The amount of free $L^{2-}$ at *t* corresponds to $I_L(t)$.

The amount of $L^{2-}$ in the $Rh_4L_4$ square at *t* corresponds to $I_{440}(t)$.

The amount of $L^{2-}$ in Int at *t* is thus expressed by $I_L(0) - I_L(t) - I_{440}(t)$.

$\langle c \rangle$

The total amount of dcb⁻ corresponds to $I_M(0)$.

The amount of free dcb⁻ at *t* corresponds to $I_{dcb}(t)$.

The amount of dcb⁻ in $Rh(dcb)_2$ at *t* corresponds to $I_M(t)$.

The amount of dcb⁻ in Int at *t* is thus expressed by $I_M(0) - I_{dcb}(t) - I_M(t)$.

The $\langle n \rangle$ and $\langle k \rangle$ values are determined with these $\langle a \rangle$, $\langle b \rangle$, and $\langle c \rangle$ values by Eqs. (1) and (2).

$$\langle n \rangle = \frac{2\langle a \rangle - \langle c \rangle}{\langle b \rangle} \quad (1)$$

$$\langle k \rangle = \frac{\langle a \rangle}{\langle b \rangle} \quad (2)$$

## NASAP of the $Rh_42_4$ square

*The reaction network model and the classification of the elementary reactions.* For the numerical analysis of self-assembly process (NASAP), a reaction network for the self-assembly of the $Rh_4L_4$ square from $RhX_2$ (X: the leaving ligand) and $L^{2-}$ ($2^{2-}$) is constructed as follows. Starting from the substrates, that is, $RhX_2$ and $L^{2-}$, the reaction path is traced to form the possible intermediate chemical species with up to ten ditopic ligands ($L^{2-}$). In this network, the $Rh_3L_3$ triangle is also considered as intermediate species. With these procedures taken, it is found that the total of 34 molecular species (including both the substrates and the product themselves) construct a reaction network composed of 1,698 elementary reactions. It should be noted that some of those reactions have the forward and backward processes (see the classification of the reaction type given below). All the intermediates considered in this network model and a simplified reaction network are shown in Supplementary Fig. 11.

This reaction network turns out to be so large that it is impossible to assign an individual rate constant to each reaction and to search for the parameter in such a vast parameter space to fit the experimental results best. Therefore, we divided the whole elementary reactions into ten classes possessing similar characteristics and defined rate constants as follows:

i. Incorporation of free $L^{2-}$ by $RhX_2$ with releasing a leaving ligand X.

$k_1$ [min⁻¹ M⁻¹] and $k_{-1}$ [min⁻¹ M⁻¹] for forward and backward reactions, respectively.

ii. Reaction of $RhX_2$ and the second site of $L^{2-}$, whose opposite site is already bonded with another $Rh$.

$k_2$ [min⁻¹ M⁻¹] and $k_{-2}$ [min⁻¹ M⁻¹].

iii. Reaction of free $L^{2-}$ and the second site of $Rh$, which is already coordinated by another $L^{2-}$.

$k_3$ [min⁻¹ M⁻¹] and $k_{-3}$ [min⁻¹ M⁻¹].

iv. Reaction at the second sites of $L^{2-}$ and $Rh$, both of whose other sites are already bonded with $Rh$ and $L^{2-}$, respectively.

$k_4$ [min⁻¹ M⁻¹] and $k_{-4}$ [min⁻¹ M⁻¹].

v. L-L exchange reaction. This reaction can lead to both the growth and the decomposition of oligomers, or produce no net change in the reaction system.

$k_5$ [min⁻¹ M⁻¹].

vi. Cyclization to make a square via L-X exchange.

$k_6$ [min⁻¹] and $k_{-6}$ [min⁻¹ M⁻¹].

vii. Cyclization to make a square via L-L exchange.

$k_7$ [min⁻¹] and $k_{-7}$ [min⁻¹ M⁻¹].

viii. Cyclization to make a triangle via L-X exchange.

$k_8$ [min⁻¹] and $k_{-8}$ [min⁻¹ M⁻¹].

ix. Cyclization to make a triangle via L-L exchange.

$k_9$ [min⁻¹] and $k_{-9}$ [min⁻¹ M⁻¹].

x. X-X exchange, which practically produces no change in the reaction system.

$k_{10}$ [min⁻¹ M⁻¹].

Typical examples of each type of elementary reaction are shown in Supplementary Fig. 12. Note that reactions in classes (i)–(v) and (x) are the intermolecular ligand exchanges and those in classes (iv)–(ix) are intramolecular ones. It should be noted here that each rate constant is defined per reaction site, based on the above modeling procedure. Therefore, the actual reaction rate for each reaction is estimated as the defined constant multiplied by the total number of available combinations. For example, for the reaction between $RhX_2$ and $L^{2-}$ to produce [$Rh$LX]⁻ and X, the rate constant is given as $k_1$ times 2 (the number of $Rh$–X bonds in $RhX_2$) times 2

(the number of coordination sites in $L^{2-}$), i.e.,

$$\boldsymbol{Rh}X_2 + L^{2-} \xrightarrow{4k_1} [\boldsymbol{Rh}LX]^- + X^-$$

It was adopted this setting to explicitly distinguish the structural difference among the species with the same composition.

*Numerical fitting of the rate constants.* In order to numerically track the time evolution of the existence ratios for both reactants and products and the $(\langle n \rangle, \langle k \rangle)$ values, we have adopted a stochastic approach based on the chemical master equation, the so-called Gillespie algorithm. In this algorithm, for all the possible $N$ chemical reactions including molecular species $S_{ai}, S_{bi}, S_{ci}, \dots$,

$$S_{ai} + S_{bi} + \cdots \rightarrow S_{ci} + \cdots (i = 1, \dots, N)$$

the total reaction rate $R_{tot}$ is calculated as

$$R_{tot} = r_1 + r_2 + \cdots + r_N (i = 1, \dots, N)$$

$$r_i = k_i [S_{ai}] [S_{bi}] \cdots$$

Starting from the initial time $t = 0$, at each instant $t$, which one of the reactions to occur is determined with the uniform random number $s_1 \in (0,1)$ as
if $s_1 \leq r_1/R_{tot}$, then reaction 1 occurs,
if $r_1/R_{tot} < s_1 \leq (r_1 + r_2)/R_{tot}$, then reaction 2 occurs,
if $(r_1 + \dots + r_{N-1})/R_{tot} < s_1 \leq 1$, then reaction N occurs.
Another uniform random number $s_2 \in (0,1)$ is independently given to fix the time incremental $dt$ as

$$dt = \ln(1/s_2)/R_{tot}$$

Time is updated as $t = t + dt$, together with the update of the numbers of corresponding molecular species, i.e., $\langle S_{ai} \rangle \rightarrow \langle S_{ai} \rangle - 1$, $\langle S_{bi} \rangle \rightarrow \langle S_{bi} \rangle - 1$, $\langle S_{ci} \rangle \rightarrow \langle S_{ci} \rangle + 1$, …. The reason why this approach traces the chemical reactions and actually works well is given in the literature in detail, along with the practical way to implement it[75–78].

The program for fixing the rate constant values is handmade for only this purpose. In order to obtain one of the best parameter sets, the following practical procedures are taken:

First, the rate constant sets are defined in the form of $k_i = 10^{m_i}$. Starting from the initial guess value for each $m_i$, time evolutions of the global self-assembly event are traced with giving each $m_i$ a stepwise increment or decrement, searching for smaller values of the residual sum of squares (RSS) for the experimental data at hand.

Initial guess sets are given in several different ways for exploring as broad parameter space as possible. An example is the uniform one like $(m_1, m_2, m_3, \dots) = (0, 0, 0, \dots)$. In another guess, rate constants for the oligomerization are given 10 to 100 times larger values than others by intention, for the reason that those reactions (especially, the inclusion of the free multitopic ligand L) generally occur very fast at the initial stage of the global self-assembly.

As a result of the global search, most of $k_i$'s settle into almost definite values, with others floating within relatively wide numerical ranges. From our numerical experiences, the latter constants do not largely affect the global reproduction of the experimental self-assembly process and the dominant reaction pathways. Therefore, for those rate constant parameters, representative values are picked up within the scope of our chemical intuition.

We have to admit that at the final point an arbitrariness occurs. And as the parameter search is performed in this procedure for giving a good fit to the experimental counterpart, we cannot

calculate the amounts of statistics such as the standard deviation for the obtained rate constant values.

With the initial conditions (numbers of species), $\langle \boldsymbol{Rh}X_2 \rangle_0 = 12,900$, $\langle \boldsymbol{2}^{2-} \rangle_0 = 12,900$, $\langle \text{others} \rangle_0 = 0$, rate constant search was performed in an eighteen-dimensional parameter space ($k_1, k_{-1}, k_2, k_{-2}, k_3, k_{-3}, k_4, k_{-4}, k_5, k_6, k_{-6}, k_7, k_{-7}, k_8, k_{-8}, k_9, k_{-9}, k_{10}$). The Avogadro number and the volume of the simulation box were set to be $N_A = 6.0 \times 10^{23}$ and $V = 2.5 \times 10^{-17}$ L, respectively, which give the same concentration as the experiments were carried out under $[\boldsymbol{2}^{2-}]_0 = [\boldsymbol{Rh}]_0 = 0.86$ mM.

Because $^1$H NMR signals of the $\boldsymbol{Rh}_3L_3$ triangle were not observed during the self-assembly (Supplementary Figs. 8 and 10) and because the conversion of the $\boldsymbol{Rh}_3L_3$ triangle into the $\boldsymbol{Rh}_4L_4$ square was slow (Supplementary Fig. 12), the rate constant values concerning the trimerization ($k_8$ and $k_9$) should be zero. Indeed, any good parameter set of the rate constants with large $k_8$ and $k_9$ was not found.

After the rate constant search was finished, refined simulations were performed for some rate parameter sets that give existence ratios and the $(\langle n \rangle, \langle k \rangle)$ plot in good agreement with the experimental counterparts. The adequacy of the fitting to the experimental data was evaluated from the residual sum of squares (RSS) with the average of the experimental data, obtained from three runs. For all the time steps $t_i$ at which the experimental data of existence ratios $R_{ex}^S$ and parameters $n_{ave}$ and $k_{ave}$ are available, RSS's are calculated with the numerically obtained values $R_{nu}^S$ as (note that $S = \boldsymbol{Rh}X_2$, $\boldsymbol{2}^{2-}$, $\boldsymbol{Rh}_4\boldsymbol{2}_4$, or X),

$$\text{RSS}_1 = \sum_{t_i} \sum_S \left( R_{nu,t_i}^S - R_{ex,t_i}^S \right)^2$$

$$\text{RSS}_2 = \sum_{t_i} \left( \langle n \rangle_{nu,t_i} - \langle n \rangle_{ex,t_i} \right)^2 + \sum_{t_i} \left( \langle k \rangle_{nu,t_i} - \langle k \rangle_{ex,t_i} \right)^2$$

A representative numerical result and the corresponding rate constant set are shown in Figs. 3d and 3e and Supplementary Table 3, respectively. In the simulation to obtain Fig. 3d, e, the initial particles and the volume of the simulation box were set to be a thousand times larger than the rough parameter search, i.e., $\langle \boldsymbol{Rh}X_2 \rangle_0 = 1,290,000$, $\langle \boldsymbol{2}^{2-} \rangle_0 = 1,290,000$, and $V = 2.5 \times 10^{-15}$ L. Although the numerical results of a single run are shown in Fig. 3d, e, a similar behavior was confirmed with several runs for the particle numbers given above.

*Determination of the major self-assembly pathway.* In the framework of the stochastic algorithm employed in this study, it is possible to count the number of occurrences (we call it as "frequency") of each elementary reaction in the course of the self-assembly process. Therefore, the dominant reaction pathway as shown in Fig. 3f was determined by tracing back the reaction network from the objective product, $\boldsymbol{Rh}_4\boldsymbol{2}_4$ in this case, to the substrates along the elementary reactions whose frequencies are larger than the others. For the elementary reaction with forward and backward processes, which direction to proceed is determined by the number difference between the forward and the backward reaction frequencies. Top 10 elementary reactions with high net frequency in the self-assembly of the $\boldsymbol{Rh}_4\boldsymbol{2}_4$ square are listed in Supplementary Table 4.

The advantage of using the reaction frequency is that the main intermediates participating in the dominant self-assembly pathway are not missed with the reason of the existence ratio being small due to high reaction rate values associated with that species. For example, the existence ratio of (4,4,1) is very low compared with other Type II intermediates (Supplementary Fig. 13, so if the major self-assembly pathway is determined by connecting the species with large existence ratio, (4,4,1) is not involved in the major pathway. On the other hand, the net reaction frequency of

the cyclization of (4,4,1) to form (4,4,0) is higher than those of the other elementary reactions to produce (4,4,0) (Supplementary Table 4), indicating that the square (4,4,0) is mainly produced by the cyclization of (4,4,1). The reason for the low existence ratio of (4,4,1) in the self-assembly is because of fast consumption of (4,4,1) by the cyclization. As this example shows, the determination of the major self-assembly pathway based on the existence ratios of the species leads to a wrong conclusion.

*X-ray crystallographic structural analysis of the [Rh₄1₄(dmso-S)₄]₂ dimer.* A single crystal was immersed in and coated with perfluoropolyalkylether (viscosity 80 cSt; abcr GmbH), and then mounted on a MicroMount™ (MiteGen LLC). Diffraction data of the single crystal were collected on a VariMax Dual single crystal X-ray diffractometer with PILATUS 200 K detector (Rigaku Corporation) at 93(2) K, using Mo $K\alpha$ ($\lambda = 0.71073$ Å) radiation monochromated by multilayer mirror optics. Bragg spots were integrated using the CrysAlisPro program package (Rigaku Corporation). An empirical absorption correction based on the multi-scan method using spherical harmonics was implemented in the SCALE3 ABSPACK scaling algorithm. The structure was solved by an intrinsic phasing method on the SHELXT program[79] and refined by a full-matrix least-squares minimization on $F^2$ executed by the SHELXL program[80], using Olex2 software package (OlexSys Ltd)[81]. Thermal displacement parameters were refined anisotropically for all non-hydrogen atoms. All the hydrogen atoms except for those on C70, C100, C137, C145 of terminal methoxy groups were located at calculated positions and the parameters were refined with a riding model. The data were corrected for disordered electron density of crystal solvents in void spaces by using the PLATON SQUEEZE method[82]. The crystal structure is shown in Fig. 4b. Crystallographic data of [**Rh₄1₄**(dmso-S)₄]₂ are summarized in Supplementary Table 7 and Supplementary Data 2.

## Data availability

The data supporting the findings of this study are available within the article and its Supplementary Information and from the corresponding author upon reasonable request. The ¹H and ¹³C{¹H} NMR spectra of all substrates are available in Supplementary Data 1. The X-ray crystallographic coordinates for the structure reported in this Article have been deposited at the Cambridge Crystallographic Data Centre (CCDC), under deposition number CCDC 2223675 ([**Rh₄1₄**(dmso-S)₄]₂). The data can be obtained with reference to Supplementary Data 2 or free of charge from The Cambridge Crystallographic Data Centre via www.ccdc.cam.ac.uk/data_request/cif.

## Code availability

All the numerical results shown in the present article were obtained with homemade program codes written in Fortran90, whose essential parts except the elementary reactions in specific network structures have already been given as the supplementary[3]. Source codes and the associated data are always available upon reasonable request to one of the authors (S.T.).

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

## Acknowledgements

This work was supported by JSPS KAKENHI grant numbers 19H02731, 19K22196, 20K05417, 21K18974, 23H01970, and 23K04663 and the Asahi Glass Foundation.

## Author contributions

S.H. conceived the project. A.O. and N.S carried out the self-assembly of the $Rh_4L_4$ squares and their characterizations by NMR measurements and QASAP. A.O. crystallized $[Rh_4\mathbf{1}_4(\text{dmso-}S)_4]_2$ and performed its X-ray analysis, and carried out a model reaction of the ligand exchange. N.S. carried out solution study on dimerization of $Rh_4\mathbf{1}_4$ and the conversion of the $Rh_3L_3$ triangles into the $Rh_4L_4$ squares. S.T. carried out NASAP and S.T. S.H., and H.S. discussed the self-assembly pathway. S.H. prepared the manuscript and all authors discussed the results and commented on the manuscript.

## Competing interests

The authors declare no competing interests.
