## [Peer Review File · Communications Chemistry]

Reviewers' comments:

Reviewer #1 (Remarks to the Author):

In this manuscript, Hiraoka and coworkers have reported that coordination squares consisting of cis-protected dinuclear Rh(II) corner complexes and linear ditopic ligands are assembled under kinetic control, perfectly preventing corresponding triangles, by modulation of their energy landscapes with a weak monotopic carboxylate ligand (dcb⁻) as a leaving ligand. The self-assembly pathway indicates that the cyclization step to form the triangular complex is blocked with dcb⁻. It was also found that dcb⁻ can convert the triangular complex into the square complex at room temperature, while heating at 373 K for 2 days is needed for the conversion without dcb⁻, indicating the catalytic effect of dcb⁻. One of the Rh(II) squares can assemble into a dimeric structure by the solvophobic effect. The work has a very good technical quality. The conclusions are well backed up by the solid experimental and numerical evidences. My feeling is that this study contains important material that should be of interest to any potential reader of Communications Chemistry. Some issues should be addressed before the final acceptance of this manuscript (see below).

1. In Fig. 2b: The yield of Rh414 is 65% as determined based on the internal standard. Which kind of species are formed for the remaining sample? Similar case exists in Fig. 2a (the total yields of Rh414 and Rh313 are 62%).
2. For supramolecular dimerization of Rh(II) square, the authors should provide the dimeric association constant on the basis of concentration- or temperature-dependent NMR measurements.

Reviewer #2 (Remarks to the Author):

In this work, the authors proposed a strategy for controlling the shape of Rh-based metal coordination assemblies. Their findings suggest that the incorporation of monotopic dcb⁻ as the leaving ligand can effectively suppress the formation of triangular assemblies, leading to the selective formation of square metal-coordination self-assemblies. Although there have been several works published on supramolecular squares with Rh²⁺ corners, there is not one focusing on the selective formation of them under kinetic control. The authors also provide experimental and numerical analysis results to reveal the mechanism for kinetically control of dcb⁻ on the self-assembly process. This manuscript offers guidance on the selective preparation of metal-coordination self-assemblies with specific shapes.

Therefore, I recommend its publication after address the minor concerns below:

1. The Mass spectrometry result is necessary to confirm the as-obtained self-assemblies. Also, it's favorable for analyzing the ratio of triangular to square complexes and tracking the conversion process of triangular to square complexes.
2. In Figure 1a, the chemical structure of the cis-protected dinuclear Rh(II) complexes shows two N-ligands, but in the following figures, the Rh²⁺ only has one N-ligand. Do they represent the same thing? I recommend the authors to unify the chemical structures and the designation of all compounds in the

diagrams (such as the use of AN and CH₃CN).

3. What does the star symbol represent in NMR spectra (Figures 2 and S6)?

4. The authors found that the addition of dcb⁻ into the mixture of R3L3 and R4L4 leads to the transition of R3L3 into R4L4. However, the authors also claimed that the coordination ability of dcb⁻ is much lower than those of the ditopic ligand. So why the monotopic dcb⁻ could replace the ditopic ligands in R3L3 since the ditopic ligands have stronger affinity to Rh?

5. The authors claimed that there is a catalytic effect of dcb⁻ in leading the conversion of R3L3 to R4L4? So, it is confused for me what is the role of dcb⁻ in selection formation of square complexes? Does dcb⁻ inhibit the formation of triangular compounds or merely promote the conversion of triangular compounds into square compounds?

6. What is the time for the fully conversion of R3L3 to R4L4 with the addition of dcb⁻? Or there will still be some triangular complex left in the mixture?

7. I think there is a mistake in the caption of Figure S2. The “Rh3L4” should be “Rh3L3”.

8. The last word in conclusion may be “field”, not “filed”.

Reviewer #3 (Remarks to the Author):

This manuscript by Hiraoka et al. describes the self-assembly based on pathway selection. This research found that the existence of Dcb as a leaving units can inhibit the formation of triangles. Moreover, dcb can quickly convert mixed assemblies into square complex. On the contrary, the conversion without dcb from triangular structure to square structure is very difficult. From an objective point of view, this work seems correctly carried out, and provides single-crystal data. The reported phenomenon by the author is interesting. However, the current trends in coordination-driven self-assembly are the precise synthesis of complex structural system and the development of new functions. As the author said “These basic knowledges would be applied to a wide range of Rh(II)-based metal-organic polyhedra (MOPs)” and “As Rh(II)-based MOPs possess high thermal and chemical stabilities and catalytic activity”, I am very interested that how to use this pathway selection to construct other MOPs, please give an example for my curiosity and show any potential properties.

Points to be addressed:

1. At the beginning I found quite muddled in Figure 1, and it took me several attempts to find what the authors were trying to convey in terms of it's novelty.
2. In Figure 1, the cartoon drawings for [Rh(AN)₆] and [Rh(dcb)₂(AN)₂] are same.
3. Also, the green ball was used for both dcb⁻ and AN.
4. Key references need to be added at the end of first paragraph in Introduction.
5. For some viewpoints, there is no conclusive evidence. For example, the Figure 3h and corresponding

description.

6. In Figure 4C, why do authors used Rh(II)-triangles?

Reviewer's comments on COMMSCHEM-23-0297-T and our responses

Reviewer 1's comments

In this manuscript, Hiraoka and coworkers have reported that coordination squares consisting of *cis*-protected dinuclear Rh(II) corner complexes and linear ditopic ligands are assembled under kinetic control, perfectly preventing corresponding triangles, by modulation of their energy landscapes with a weak monotopic carboxylate ligand (dcb^-) as a leaving ligand. The self-assembly pathway indicates that the cyclization step to form the triangular complex is blocked with dcb^- . It was also found that dcb^- can convert the triangular complex into the square complex at room temperature, while heating at 373 K for 2 days is needed for the conversion without dcb^- , indicating the catalytic effect of dcb^- . One of the Rh(II) squares can assemble into a dimeric structure by the solvophobic effect. The work has a very good technical quality. The conclusions are well backed up by the solid experimental and numerical evidences. My feeling is that this study contains important material that should be of interest to any potential reader of Communications Chemistry. Some issues should be addressed before the final acceptance of this manuscript (see below).

1. In Fig. 2b: The yield of Rh_4I_4 is 65% as determined based on the internal standard. Which kind of species are formed for the remaining sample? Similar case exists in Fig. 2a (the total yields of Rh_4I_4 and Rh_3I_3 are 62%).
2. For supramolecular dimerization of Rh(II) square, the authors should provide the dimeric association constant on the basis of concentration- or temperature-dependent NMR measurements.

Reviewer 1's Comment 1

In Fig. 2b: The yield of Rh_4I_4 is 65% as determined based on the internal standard. Which kind of species are formed for the remaining sample?

Our response

In the ^1H NMR spectrum measured after the convergence of the self-assembly, signals other than those of the ring(s) and the leaving ligand (dcb^-) were almost not observed. Thus, the species produced after self-assembly, which would be intermediates or kinetic traps, could not be characterized by NMR spectroscopy. According to QASAP (n - k analysis), the (n , k) value after the convergence indicates that Type II species consisting of 4 or 5 Rh dinuclear units on average are suggested. As the ^1H NMR signals of such linear species were not observed, a mixture of type II species with a different chain length would be produced and/or their NMR signals would broaden, probably due to dynamic motion.

Reviewer 1's Comment 2

For supramolecular dimerization of Rh(II) square, the authors should provide the dimeric association constant on the basis of concentration- or temperature-dependent NMR measurements.

Our response

According to the reviewer's suggestion, the dimerization constant was evaluated using concentration-

dependent ^1H NMR measurements. The ^1H NMR signals assigned to the monomer square (Rh_4L_4) did not appear even at a concentration of $4.6\ \mu\text{M}$ ($[Rh_4L_4]_0 = 4.6\ \mu\text{M}$). Further dilution of the solution caused strong broadening and weak intensity of the NMR signals, which did not allow us to determine whether the monomer signals appeared. Therefore, although the exact dimerization constant was not determined, it is estimated to be higher than $10^7\ \text{M}^{-1}$ based on the result of no ^1H NMR signals of the monomer at $4.6\ \mu\text{M}$ under the assumption that 90% of the monomer dimerizes in solution; 10% monomer square exists. This result has been added to the main text with Fig. S20 in the revised manuscript and Supplementary Information.

Reviewer 2's comments

In this work, the authors proposed a strategy for controlling the shape of Rh-based metal coordination assemblies. Their findings suggest that the incorporation of monotopic dcb^- as the leaving ligand can effectively suppress the formation of triangular assemblies, leading to the selective formation of square metal-coordination self-assemblies. Although there have been several works published on supramolecular squares with Rh^{2+} corners, there is not one focusing on the selective formation of them under kinetic control. The authors also provide experimental and numerical analysis results to reveal the mechanism for kinetically control of dcb^- on the self-assembly process. This manuscript offers guidance on the selective preparation of metal-coordination self-assemblies with specific shapes. Therefore, I recommend its publication after address the minor concerns below:

1. The Mass spectrometry result is necessary to confirm the as-obtained self-assemblies. Also, it's favorable for analyzing the ratio of triangular to square complexes and tracking the conversion process of triangular to square complexes.
2. In Figure 1a, the chemical structure of the *cis*-protected dinuclear Rh(II) complexes shows two N-ligands, but in the following figures, the Rh^{2+} only has one N-ligand. Do they represent the same thing? I recommend the authors to unify the chemical structures and the designation of all compounds in the diagrams (such as the use of AN and CH_3CN).
3. What does the star symbol represent in NMR spectra (Figures 2 and S6)?
4. The authors found that the addition of dcb^- into the mixture of Rh_3L_3 and Rh_4L_4 leads to the transition of Rh_3L_3 into Rh_4L_4 . However, the authors also claimed that the coordination ability of dcb^- is much lower than those of the ditopic ligand. So why the monotopic dcb^- could replace the ditopic ligands in Rh_3L_3 since the ditopic ligands have stronger affinity to Rh?
5. The authors claimed that there is a catalytic effect of dcb^- in leading the conversion of Rh_3L_3 to Rh_4L_4 ? So, it is confused for me what is the role of dcb^- in selection formation of square complexes? Does dcb^- inhibit the formation of triangular compounds or merely promote the conversion of triangular compounds into square compounds?
6. What is the time for the fully conversion of Rh_3L_3 to Rh_4L_4 with the addition of dcb^- ? Or there will still be some triangular complex left in the mixture?
7. I think there is a mistake in the caption of Figure S2. The " Rh_3L_4 " should be " Rh_3L_3 ".

8. The last word in conclusion may be “field”, not “filed”.

Reviewer 2's comment 1

The Mass spectrometry result is necessary to confirm the as-obtained self-assemblies. Also, it's favorable for analyzing the ratio of triangular to square complexes and tracking the conversion process of triangular to square complexes.

Our Response

According to the reviewer's kind suggestion, we attempted to measure the ESI-TOF mass spectrometry of the squares, triangles, and their mixtures from the ditopic ligand **1** and from **2** and successfully detected their signals. These results are shown in Figure S4 in the revised Supplementary Information. As to the mixture of the Rh_3L_3 triangle and the Rh_4L_4 square ($L = 1^{2-}$ or 2^{2-}), the signals for them were detected by mass spectrometry, while the solution obtained after the conversion of the Rh_3L_3 triangles into the Rh_4L_4 squares upon addition of dcb^- showed only the signals for the Rh_4L_4 squares, which can support the assignment of the square and triangle by mass spectrometry. The mass signals of the triangles and squares are not strong probably because the complexes are electronically neutral. The mass signals of the triangles are relatively weak compared with those of the squares. One of the reasons for the weak signals of the triangles would be the decomposition of the metastable triangles during the ionization process.

Regarding the monitoring of the conversion of the triangles into the squares promoted by dcb^- , the change in the ratio of the Rh_3L_3 triangle to the Rh_4L_4 square complexes for ditopic ligands 1^{2-} and 2^{2-} with time is summarized in Tables S5 and S6 in the revised Supplementary Information and these tables are cited from the main text.

Reviewer 2's comment 2

In Figure 1a, the chemical structure of the *cis*-protected dinuclear Rh(II) complexes shows two N-ligands, but in the following figures, the Rh^{2+} only has one N-ligand. Do they represent the same thing? I recommend the authors to unify the chemical structures and the designation of all compounds in the diagrams (such as the use of AN and CH_3CN).

Our response

Thank you for this reviewer's kind advice regarding the indication of the chemical structures. In Figures 2 and 4c, partial chemical structures are shown for assigning the 1H NMR signals. To avoid confusion, the chemical structure of the *cis*-protected ligand is shown in the revised Figures 2 and 4c. Additionally, we have revised Figure 1, where the designation of acetonitrile is unified as CH_3CN for clarity.

Reviewer 2's comment 3

What does the star symbol represent in NMR spectra (Figures 2 and S6)?

Our response

The asterisks shown in the 1H NMR spectra in Figures 2 and S6 (S7 in the revised SI)

at approximately 7.2 ppm represent the carbon satellite of CHCl_3 . This information is indicated in the captions of these figures in the revised manuscript and Supplementary Information.

Reviewer 2's comment 4

The authors found that the addition of dcb^- into the mixture of Rh_3L_3 and Rh_4L_4 leads to the transition of Rh_3L_3 into Rh_4L_4 . However, the authors also claimed that the coordination ability of dcb^- is much lower than those of the ditopic ligand. So why the monotopic dcb^- could replace the ditopic ligands in Rh_3L_3 since the ditopic ligands have stronger affinity to Rh?

Our response

When ligand exchange occurs through an associative process on the metal center (Pd(II), Pt(II), and Rh(II)), weak ligands promote the ligand exchange of stronger ligands. For example, for Pd(II) complexes, a coordinative solvent (CH_3CN and DMSO) and a coordinative counter anion (NO_3^-) accelerate the ligand exchanges of the N-donor ligand (L), such as pyridine, even though the coordination ability of CH_3CN , DMSO, and NO_3^- is weaker than that of L. The catalytic effect of dcb^- found in this research is similar to the role of CH_3CN , DMSO, and NO_3^- for Pd(II) complexes. Another reason for the conversion of Rh_3L_3 into Rh_4L_4 in the presence of dcb^- would be due to the distortion around the Rh(II) centers in Rh_3L_3 , which makes breaking the Rh(II)–L bond by dcb^- in Rh_3L_3 more easily than in other Rh(II)-based complexes without such a distortion.

Reviewer 2's comment 5

The authors claimed that there is a catalytic effect of dcb^- in leading the conversion of Rh_3L_3 to Rh_4L_4 ? So, it is confused for me what is the role of dcb^- in selection formation of square complexes? Does dcb^- inhibit the formation of triangular compounds or merely promote the conversion of triangular compounds into square compounds?

Our response

As pointed by the reviewer, we revealed the catalytic effect of dcb^- in leading the conversion of Rh_3L_3 to Rh_4L_4 ; however, it hardly influences the selective formation of the square complexes, because the time scale of the conversion of the triangle into the square with dcb^- is much slower than that of the assembly of the square from the substrates ($\text{Rh}(\text{dcb})_2$ and the ditopic ligand). To avoid confusion, we add to the revised manuscript the following sentences: “However, the time scale of the conversion of the triangle into the square using the catalyst is much slower than that of the assembly of the square from the substrates ($\text{Rh}(\text{dcb})_2$ and the ditopic ligand). The idea that the selective self-assembly of the Rh_4L_4 squares took place by the conversion of the Rh_3L_3 triangles into the Rh_4L_4 squares by the catalytic effect of dcb^- during the self-assembly is ruled out by the fact that the signals of the Rh_3L_3 triangles were not observed during the self-assembly (Figs. 2c and S8)”.

The dominant role of dcb^- in the selective self-assembly is to prevent the cyclization of the $\text{Rh}_3\text{L}_3(\text{dcb})$ chain intermediate. This sentence has been described in Conclusions.

Reviewer 2's comment 6

What is the time for the fully conversion of Rh_3L_3 to Rh_4L_4 with the addition of dcb^- ? Or there will still

be some triangular complex left in the mixture?

Our response

It took approximately two days for the conversion of the triangles into squares at room temperature in the presence of dcb^- . According to the ^1H NMR obtained after 2 days, less than 4% of the triangle was left in the solution. The existence ratios of the species were determined based on the internal standard.

Reviewer 2's comment 7

I think there is a mistake in the caption of Figure S2. The " Rh_3L_4 " should be " Rh_3L_3 ".

Our response

Thank you for letting us know about this mistake. The caption in Figure S2 has been revised.

Reviewer 2's comment 8

The last word in conclusion may be "field", not "filed".

Our response

Thank you for highlighting the typographical errors in the manuscript. The typo has been corrected in the revised manuscript.

Reviewer 3's comments

This manuscript by Hiraoka *et al.* describes the self-assembly based on pathway selection. This research found that the existence of dcb^- as a leaving units can inhibit the formation of triangles. Moreover, dcb^- can quickly convert mixed assemblies into square complex. On the contrary, the conversion without dcb^- from triangular structure to square structure is very difficult. From an objective point of view, this work seems correctly carried out, and provides single-crystal data. The reported phenomenon by the author is interesting. However, the current trends in coordination-driven self-assembly are the precise synthesis of complex structural system and the development of new functions. As the author said "These basic knowledges would be applied to a wide range of Rh(II)-based metal-organic polyhedra (MOPs)" and "As Rh(II)-based MOPs possess high thermal and chemical stabilities and catalytic activity", *I am very interested that how to use this pathway selection to construct other MOPs, please give an example for my curiosity and show any potential properties.* Points to be addressed:

2. At the beginning I found quite muddled in Figure 1, and it took me several attempts to find what the authors were trying to convey in terms of it's novelty.
3. In Figure 1, the cartoon drawings for $[\text{Rh}(\text{AN})_6]$ and $[\text{Rh}(\text{dcb})_2(\text{AN})_2]$ are same.
4. Also, the green ball was used for both dcb^- and AN
5. Key references need to be added at the end of first paragraph in Introduction.
6. For some viewpoints, there is no conclusive evidence. For example, the Figure 3h and corresponding description.
7. In Figure 4C, why do authors used Rh(II)-triangles?

Reviewer 3's comment 1

I am very interested that how to use this pathway selection to construct other MOPs, please give an example for my curiosity and show any potential properties.

Our response

Considering that the dinuclear Rh(II) complex is geometrically the same as the square planar Pd(II) ion center, it is expected that geometrically similar metal-organic cages reported based on Pd(II) ions and multitopic N-ligands can be realized from dinuclear Rh(II) centers and multitopic carboxylate ligands. However, the examples for Rh(II)-based metal-organic cages are much fewer than those for Pd(II)-based cages. Interestingly, the dinuclear Mo(II) center, which is the isostructure of the dinuclear Rh(II) center, provides several structures of metal-organic cages (J.-R. Li *et al.*, *J. Am. Chem. Soc.*, **2010**, *132*, 17599, DOI: 10.1021/ja1080794; Y. Ke *et al.*, *Inorg. Chem.* **2005**, *44*, 4154, DOI: 10.1021/ic050460z). In general, the main reason for the high efficiency of molecular self-assembly stems from error correction due to the reversibility of the interactions between the building blocks. Under these conditions, the thermodynamically most stable assembly is produced. Therefore, the success of Mo(II)-based metal-organic cages is mainly because the ligand exchange of Mo(II)-carboxylate is much faster than that of Rh(II)-carboxylate. Unfortunately, Mo(II)-based metal-organic cages are unstable in air; therefore, the application of these complexes is limited. Because the Rh(II)-carboxylate bond is more inert than the Pd(II)-N coordination bond, it is difficult to achieve equilibration in Rh(II)-based self-assembly, co-producing kinetically trapped species. In previous research, heating at high temperatures for a long time has been applied to solve this problem.

The results of this research for the *cis*-protected dinuclear Rh(II) complex-based square self-assembly are as follows: (1) the leaving ligand affects the self-assembly pathway, altering the energy barriers of certain steps, and (2) relatively slow ligand exchange of Rh-carboxylate bond can be accelerated by weak monocarboxylate such as dcb^- . These findings can be used for Rh(II)-based self-assembly from thermodynamic and kinetic viewpoints. In thermodynamically controlled molecular self-assembly, the acceleration of ligand exchanges in Rh(II)-carboxylate bonds enables Rh(II)-based self-assembly to reach equilibration, efficiently leading to thermodynamically stable assemblies. The acceleration of the ligand exchanges can be tuned by the choice of the leaving ligand. On the other hand, the advantage of kinetically controlled molecular self-assembly is that metastable assemblies can be obtained as a major species when the self-assembly pathway to such species is selected. Because Rh(II)-carboxylate bonds are stronger than Pd(II)-N bonds, the stability of metastable Rh(II)-based self-assemblies is higher than those of Pd(II)-based ones. Therefore, the applicability of such metastable assemblies is wider. As observed in the selective formation of Rh_4L_4 squares in this study, certain elementary steps are blocked by the choice of the leaving ligand. Similarly, the pathway selection caused by the leaving ligand would enable the selective production of metastable Rh(II)-based assemblies. Accordingly, it is expected that a proper choice of monocarboxylate as the leaving ligand and/or catalysis under thermodynamic and kinetic control enables us to create thermodynamically stable and metastable Rh(II)-based metal-organic cages.

Reviewer 3's comment 2

At the beginning I found quite muddled in Figure 1, and it took me several attempts to find what the authors were trying to convey in terms of its novelty.

Our response

Thank you for this reviewer's kind advice on improving Figure 1. Figure 1 has been completely edited in the revised manuscript so that readers can easily understand the point of this research.

Reviewer 3's comments 3 and 4

In Figure 1, the cartoon drawings for $[Rh(AN)_6]$ and $[Rh(dcb)_2(AN)_2]$ are same. Also, the green ball was used for both dcb^- and AN.

Our response

According to the reviewer's kind advice, the cartoon representations of $[Rh(AN)_6]$ and $[Rh(dcb)_2(AN)_2]$ have been changed in the revised Figure 1.

Reviewer 3's comment 5

Key references need to be added at the end of first paragraph in Introduction.

Our response

The reviewer kindly suggests that references concerning "a general principle of pathway selection in complicated reversible reaction networks" should be cited in the Introduction section. Although examples of pathway selection in reversible reaction networks, such as molecular self-assembly, have been reported, the "general principle" has not yet been established. Recently, we reported a general principle in a reversible reaction network (Takahasi, S. *et al.*, *Chem*, DOI: 10.1016/j.chempr.2023.06.015). In this paper as well, we state the following: "an understanding of how kinetic traps are produced is elusive because the general principle of pathway selection in reversible reaction networks has not been established yet." without references. Therefore, the above-mentioned recent paper is cited at the end of the introduction in the revised manuscript.

Reviewer 3's comment 6

For some viewpoints, there is no conclusive evidence. For example, the Figure 3h and corresponding description.

Our response

Figure 3h shows the proposed structure of the intermediate or transition state for triangle formation. According to the experimental results and numerical analysis of the experimental data obtained by QASAP, the cyclization of linear species to form the triangle (Rh_3L_3) is prevented except in the case where the metal source is $Rh(CH_3CN)_4$. Therefore, it is expected that a small leaving ligand (CH_3CN) enables triangle formation, while large leaving ligands such as dcb^- for $Rh_3L_3(dcb)$ and $Rh_{x+1}L_{x+1}(dcb)$ for the Type II intermediates ($Rh_{m+x}L_{m+x}(dcb)$) shown in Figure 3g and 3h prevent triangle formation, probably due to the steric effect around the large leaving ligand. This discussion is summarized in the caption of Figure 3h in the revised manuscript.

Reviewer 3's comment 7

In Figure 4C, why do authors used Rh(II)-triangles?.

Our response

The cartoon representation in Figure 4c in the original manuscript is a partial structure of the dimer of the ***Rh*₄1₄** square so that the readers can understand the assignment of the ¹H NMR signals. Based on the reviewer's comment on this cartoon, to avoid confusion, this partial structure has been removed from Figure 4c in the revised manuscript.

REVIEWERS' COMMENTS:

Reviewer #1 (Remarks to the Author):

The authors have addressed all of the issues raised by the reviewers. I recommend the publication of this manuscript in its current form.

Reviewer #2 (Remarks to the Author):

I am satisfied with the revision.

Reviewer #3 (Remarks to the Author):

[Editorial Note: This reviewer provided no further comments for the authors.]

Reviewer's comments on COMMSCHEM-23-0297-T and our responses

Reviewer 1's comments

The authors have addressed all of the issues raised by the reviewers. I recommend the publication of this manuscript in its current form.

Our response

Thank you very much for this reviewer's kind review of our manuscript and giving us valuable comments in the previous review. We are happy to hear that the reviewer recommended the publication the revised manuscript.

Reviewer 2's comments

I am satisfied with the revision.

Our response

Thank you very much for this reviewer's kind review of our manuscript and giving us valuable comments in the previous review. We are happy to hear that the current version of manuscript satisfied the reviewer.

Reviewer 3's comments

[Editorial Note: This reviewer provided no further comments for the authors.]

Our response

Thank you very much for this reviewer's kind review of our manuscript and giving us valuable comments in the previous review.